# A context-free model of savings in motor learning

**Mahdiyar Shahbazi[1]‡, Olivier Codol[2,3], Jonathan A Michaels[4,5]†, Paul L Gribble[1,4]\*†**

[1]Department of Psychology, Western University, London, Canada; [2]Mila–Québec Artificial Intelligence Institute, Montréal, Canada; [3]Department of Neuroscience, Université de Montréal, Montréal, Canada; [4]Department of Physiology and Pharmacology, Schulich School of Medicine and Dentistry, London, Canada; [5]School of Kinesiology and Health Science, Faculty of Health, York University, Toronto, Canada

**\*For correspondence:**
pgribble@uwo.ca

†These authors contributed equally to this work

**Present address:** ‡Dept. Organismic and Evolutionary Biology, Harvard University, Boston, United States

**Competing interest:** The authors declare that no competing interests exist.

## eLife Assessment

This study presents **valuable** computational findings on the neural basis of learning new motor memories and the savings using recurrent neural networks. The evidence supporting the claims of the authors is **solid**, but it would benefit from more detailed discussion on the specific conditions under which savings emerges from purely implicit mechanisms. This work will be of interest to computational and experimental neuroscientists working in motor learning.

**Abstract** Learning to adapt voluntary movements to an external perturbation, whether mechanical or visual, is faster during a second encounter than during the first. The mechanisms underlying this phenomenon, known as savings, remain unclear. Recent studies propose that the high dimensionality of neural control enables the retention of learning traces that may facilitate savings. To test this idea, we used MotorNet, a framework for training recurrent neural networks (RNNs) to control biomechanical models of the human upper limb. RNNs were trained to perform reaching movements with a velocity-dependent force field (FF) and without (NF) in the sequence NF1 (baseline), FF1 (adaptation), NF2 (washout), and FF2 (re-adaptation). RNNs showed behavioural signatures of savings in the absence of any explicit contextual input signalling the presence or absence of the FF. Savings was more robust in RNNs with larger numbers of units. We identified a component of RNN activity associated with savings—a shift in preparatory activity that persisted even after washout. Displacing this preparatory activity in the direction of the shift enhanced savings, whereas perturbations in the opposite direction reduced or eliminated savings. These findings suggest a potential neural basis for motor memory retention underlying savings that is reliant on the high dimensionality of neural circuits for control, and is independent of cognitive or strategic learning.

## Introduction

In studies of motor learning, *savings* commonly refers to a phenomenon in which learning is superior after previous exposure to a motor task. Savings has been demonstrated in the context of voluntary reaching movements for adaptation to novel visuomotor perturbations and for learning to counter novel mechanical environments, such as velocity-dependent curl force fields (FF) (*Haith et al., 2015*; *Morehead et al., 2015*; *Leow et al., 2016*; *Nguyen et al., 2019*; *Yin and Wei, 2020*; *Hadjiosif et al., 2023*; *Coltman et al., 2019*). In a typical experiment, participants are initially exposed to a novel perturbation environment and they practice reaching to targets until an asymptotic level of performance is reached, for example, recovery of approximately straight-line hand paths. Following this

initial learning, the perturbation is removed, and participants practice again until behavioral performance in this 'washout' phase returns to pre-learning baseline levels. After washout participants are re-exposed to the perturbing environment. Savings is observed as a faster learning rate when re-exposed to the perturbing environment compared to initial learning, and sometimes also as a superior initial level of performance compared to when the perturbing environment was first encountered (*Coltman et al., 2019*; *Herzfeld et al., 2014*).

There is some debate about the mechanisms that may be responsible for savings (*Leow et al., 2016*; *Yin and Wei, 2020*). Some propose that savings is produced by explicit, cognitive, or strategic processes, such as a conscious memory of the action selection strategy (*Morehead et al., 2015*), a contextual signal associated with the perturbing environment (*Heald et al., 2021*), meta-learning of adaptation parameters, such as learning rate (*Zarahn et al., 2008*; *McDougle et al., 2015*; *Albert and Shadmehr, 2018*), or reinforcement-based memories of successful execution (*Huang et al., 2011*). Others have proposed that savings may arise from implicit learning processes that are not based on conscious, strategic mechanisms, including an increased sensitivity to previously experienced errors (*Herzfeld et al., 2014*), use-dependent plasticity (*Diedrichsen et al., 2010*), or implicit updating of internal models that predict the sensory consequences of action (*Wolpert et al., 1995*).

Recent advances have been made in the ability to record from large numbers of neurons during motor learning tasks (*Trautmann et al., 2025*) and this has resulted in new approaches to understanding the relationship between high dimensional neural population activity and sensory, motor, and task parameters during, and even prior to, voluntary movement (*Kobak et al., 2016*; *Dubreuil et al., 2022*). Sun, O'Shea, and colleagues recorded neural activity in primary motor cortex of rhesus macaques during a FF reaching task and identified a neural subspace of network activity during the preparatory period prior to movement that shifted after learning (*Sun et al., 2022*). This 'uniform shift' persisted even after behavioral washout of FF learning. The authors proposed that this neural trace of prior learning could facilitate subsequent savings (*Sun et al., 2022*). Losey and colleagues used a brain-computer interface learning paradigm to study how neural population activity in the primary motor cortex of monkeys supports motor learning of multiple tasks (*Losey et al., 2024*). They identified a change in a subspace of neural population activity that supported behavioral performance of a new task without interfering with a previously learned task. They proposed that the high dimensionality of neural activity in primary motor cortex allows for the formation of memory traces that supports multiple behaviors without interference.

In the present paper, we used artificial recurrent neural network (RNN) models of upper limb motor control to test the idea that high-dimensional neural control facilitates the encoding of multiple novel motor memories, and that neural traces of previous learning underlie subsequent savings, without the need for contextual signals. We used MotorNet (*Codol et al., 2024b*) to train RNNs to control a mathematical model of the upper limb neuromuscular system (*Kistemaker et al., 2010*) in the context of a simulated FF reaching task (*Shadmehr and Mussa-Ivaldi, 1994*; *Conditt et al., 1997*). Even without any explicit contextual cue signalling the presence of absence of FFs, RNNs showed behavioral signatures of savings. In addition, savings was more robust as the number of units in RNNs increased. Using approaches similar to those described in previous studies we identified a learning-related shift in neural activity in the preparatory period prior to movement onset (*Sun et al., 2022*; *Losey et al., 2024*). We established a causal relationship between this neural shift and savings by perturbing neural activity along the direction of this neural shift. When RNN hidden unit activity was shifted in the direction of the neural shift, savings was enhanced, whereas neural perturbations in the opposite direction reduced or abolished savings. Our findings support the hypothesis that a neural basis of motor memory retention underlies savings, one that could be independent of cognitive or strategic learning components and that depends upon the high dimensionality of neural population activity.

## Results

We trained 40 recurrent neural networks (RNNs) with 128 fully connected gated recurrent units (GRUs) to control a mathematical model of the human upper limb (*Codol et al., 2024b*; *Figure 1a and b*). Task inputs to the RNN are the Cartesian coordinates of the movement target $(x, y)$ and a binary go signal (0 or 1) indicating when to initiate movement (*Figure 1c*). The RNN also receives time-delayed feedback signals corresponding to the length and velocity of each muscle, and the Cartesian coordinates of the endpoint of the limb. The output of the RNN is time-varying muscle stimulation

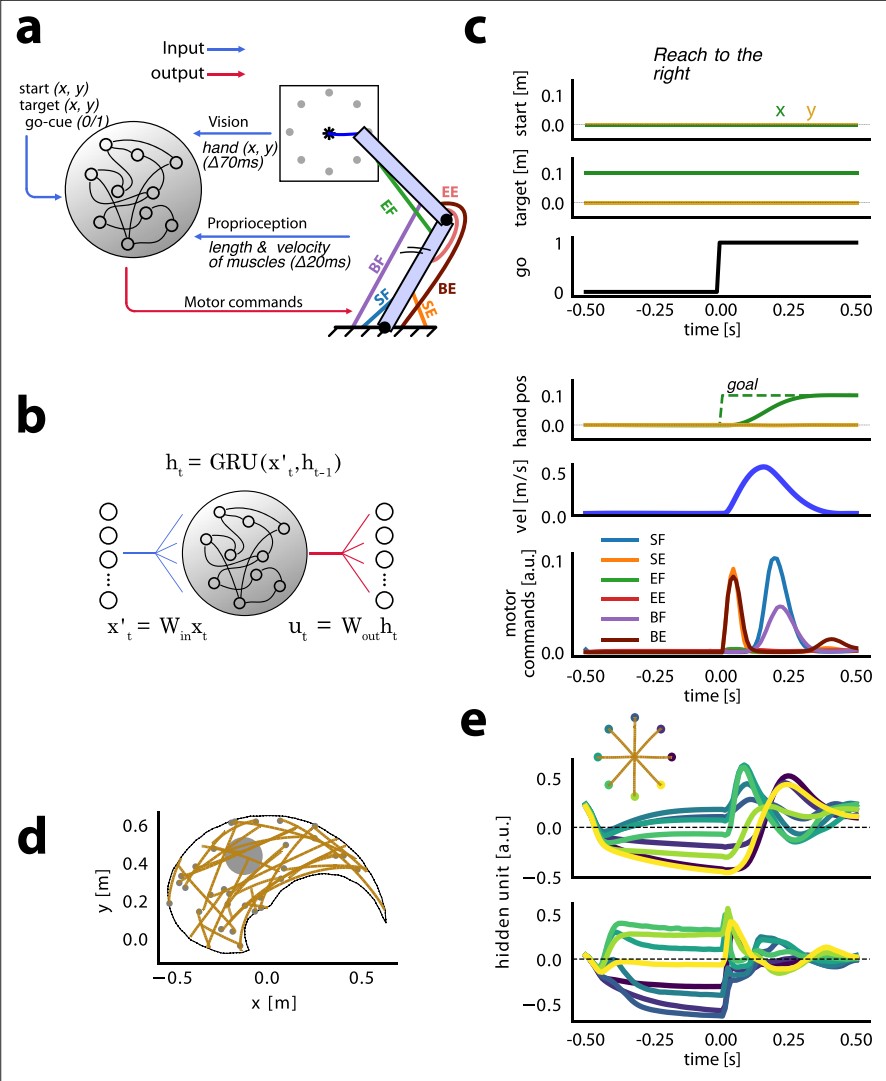

**Figure 1.** Recurrent neural network model inputs and outputs. (**a**) Recurrent neural networks (RNNs) receive a 17-dimensional input signal consisting of the location of the movement target in Cartesian coordinates, a 'visual' feedback signal giving the arm's endpoint position delivered with a 70 ms delay, a 'proprioceptive' feedback signal consisting of the length and velocity of each of the 6 limb muscles delivered with a 20 ms delay, and a binary go cue. RNNs output 6 motor stimulation commands (between 0 and 1) to drive each muscle: SF (shoulder flexors), SE (shoulder extensors), EE (elbow extensors), EF (elbow flexors), BE (bi-articular extensors), and BF (bi-articular flexors). (**b**) The 17-dimensional input signal was mapped to the recurrent network using linear weights $W_{in}$. RNN output was transformed into motor commands by linear weights $W_{out}$. The vector $h_t$ is the activity of hidden units at time $t$. (**c**) Task-related RNN inputs for a reach in a null field toward the rightmost target depicted in (**a**). For the purpose of illustration in this Figure, we translated the starting and target positions such that the start position is at the coordinates (0, 0). The simulation duration was 1 s, with 10ms time steps. The goal (dashed lines) was set to the hand's starting position before the go signal changed to 1, to the movement target position after that. (**d**) Sample endpoint trajectories after training RNNs on reaches to random target locations. Reaching trajectories are indicated in orange, and small gray dots show target positions. The large gray circle indicates the position of the centre-out reaches within the workspace. (**e**) Reaching trajectories and hidden unit activity from two example hidden units over time at the end of training in the centre-out task. colors indicate each of eight targets. The go-cue switches from 0 to 1 at time $t = 0$.

commands to each of 6 upper limb muscles (*Figure 1c*). Muscle stimulation commands range between 0 and 1 and act on a musculoskeletal model of the upper limb which includes multi-joint limb dynamics and a hill-type muscle model (*Kistemaker et al., 2010*).

The networks were initially trained to produce point-to-point reaching movements between targets located in random locations throughout the limb's workspace. No perturbing FF was applied during this initial 'growing up' training phase. We refer to the absence of a perturbing FF as a 'null field' (NF). RNN weights were updated using backpropagation through time (*Werbos, 1990*), using the Adam optimizer (*Kingma and Ba, 2014*) implemented in PyTorch (*Paszke et al., 2019*). The loss function for optimization was based on minimizing the difference between hand position and target position, and also included regularization terms that encouraged the network to produce smooth, human-like kinematics, phasic muscle commands, and stable hidden unit activity (*Michaels et al., 2020*; *Sussillo et al., 2015*) (see *Figure 1e* and Methods for details).

After training, the networks produced reaching movements with human-like characteristics, including smooth, relatively straight hand paths with bell-shaped velocity profiles, and phasic activity in agonist and antagonist muscles spanning the shoulder and elbow (*Figure 1c*).

*Figure 1d* shows examples of reaching trajectories for reaches to targets located randomly across the limb's workspace. Models produced human-like reaches both when tested on reaches with random starting points and targets (*Figure 1d*) and when tested on centre-out reaches toward 8 equidistant targets (*Figure 1e*). Consistent with electrophysiological recordings in monkeys, RNN hidden units showed activity patterns that were relatively stable over time, and distinct for different movement targets during the delay period prior to the go signal (time 0 in *Figure 1e*). RNN hidden unit activity during movement was similarly distinct for different movement directions, and showed oscillatory activity consistent with that seen in recordings from motor cortex in non-human primates (*Churchland et al., 2012*; *Churchland and Shenoy, 2024*) (also see Figure 6).

## Force field adaptation

After the networks were trained to perform point-to-point reaches in a NF, we implemented a relatively standard experimental sequence common in studies of human motor learning. We trained networks on a centre-out reaching task either in the absence of perturbing forces (null field, NF) or in the presence of a velocity-dependent curl force field (FF) (see Methods). First, networks were exposed to a NF (NF1) to characterize baseline performance. Following this, networks were trained to produce reaches in a clockwise FF (FF1, 3200 batches of training). After initial FF learning networks were re-trained in a NF (NF2, 10,000 batches). Following this 'washout' phase networks were re-trained in the FF (FF2, 3200 batches) (see *Figure 2a*).

We characterized behavioral performance of the networks by measuring the maximum deviation of each hand trajectory from a straight line connecting initial and final target positions. During centre-out reaching in the initial NF baseline tests (NF1, *Figure 2b*), the network produced straight hand paths with very little lateral deviation.

We then trained the networks to compensate for the effects of a clockwise force field (FF1). The first time networks encountered the FF (batch 0), they exhibited large lateral deviations from a straight line trajectory (*Figure 2a and b*). Over the course of training, the network's hidden weights were modified so that the networks produced different muscle stimulation commands that compensated for the forces produced by the FF (*Figure 2b*). Only the hidden (recurrent) weights were modified after the 'growing up' phase, and not the input/output weights. By the end of FF1, relatively straight hand paths were recovered, and lateral deviation was near that in the NF baseline tests (NF1).

Importantly, at no time during training did the networks receive any contextual signal related to the presence or absence of the FF. Adaptation occurred during FF training because as the simulated limb is perturbed by the FF, hand position deviates away from the target, and the loss function increases. Over training the values of the RNN hidden unit weights are changed to minimize the loss function, and in turn, recover straight hand paths.

## Washout

After force field adaptation (FF1) we trained the networks in a washout phase (NF2) in which we removed the simulated FF that the networks trained on in FF1. As is the case in empirical studies of FF learning, the networks initially showed an after-effect in movement kinematics in the opposite

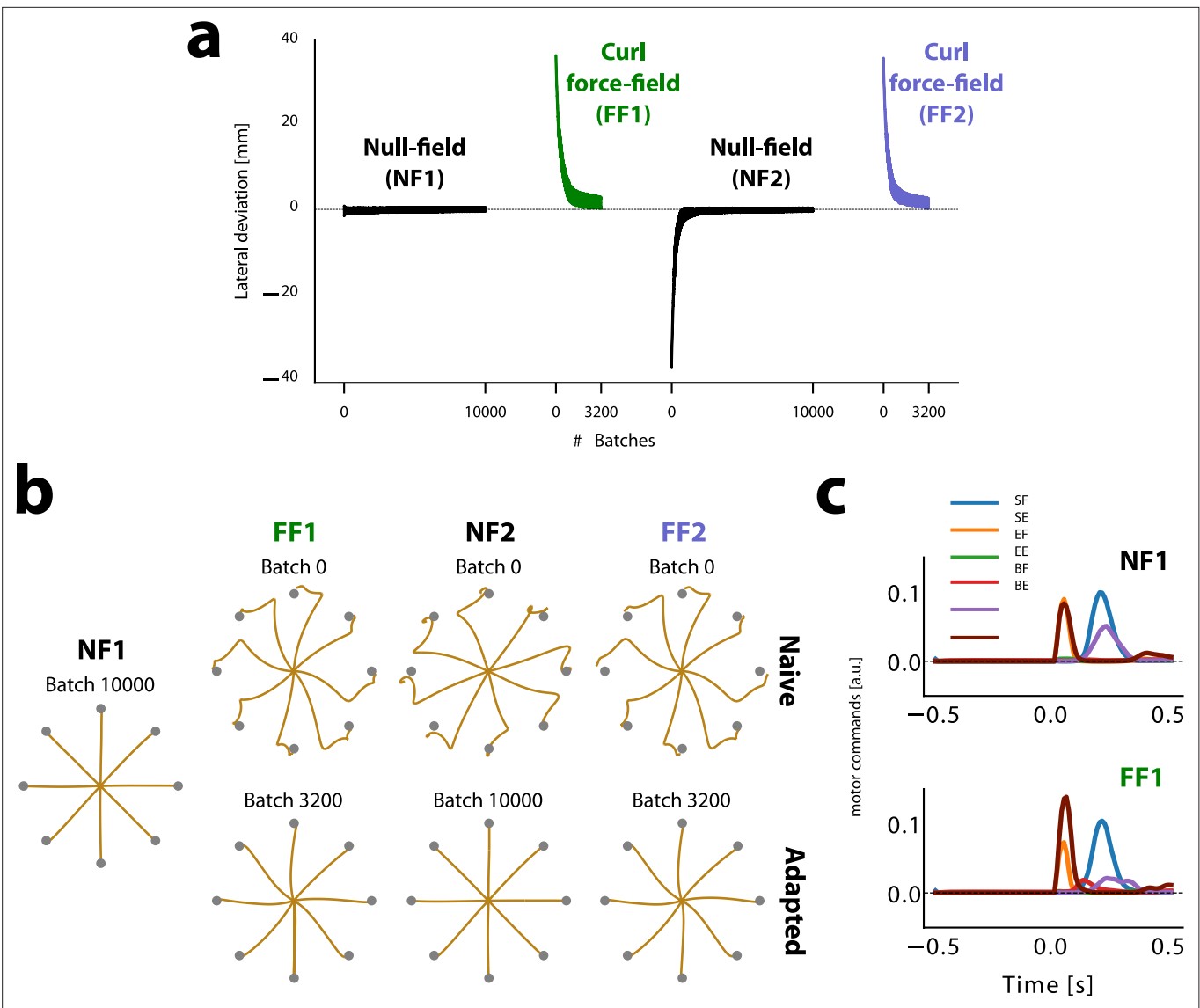

**Figure 2.** Networks learn to compensate for curl force fields without any contextual input. (**a**) Lateral deviation averaged over 8 centre-out reaches for each batch. Black indicates null-field phases (NF1 and NF2), green indicates the first phase of the force field (FF1), and purple indicates the second phase of the force field (FF2). Positive values indicate deviation in the direction of the force field, which is clockwise relative to the line connecting the starting and target positions. (**b**) Simulated reaching trajectories at the beginning and end of each phase, grouped in different columns. (**c**) Motor commands during reaching toward the rightmost target for NF1 and FF1.

direction of the force field (*Figure 2a and b*), indicating that the networks prepared muscle commands to compensate for the (now absent) FF. After training in NF2, networks recovered straight line hand paths that were very similar to the performance in the NF1 baseline phase (only 0.1 mm difference in lateral deviation).

## Re-adaptation

Following washout, we re-trained networks in the same curl field (FF2) that they had been exposed to previously in the FF1 block. We observed that when networks initially encountered the FF in FF2, they exhibited smaller lateral deviations than those observed in the beginning of FF1, when they were first exposed to the simulated FF ($t(39) = 10.940$, $p = 1.9e - 13$) (*Figure 3a*). In addition, networks adapted to the force field in FF2 faster than they did in FF1, as measured by an increased learning rate based on an exponential fit to the learning curve (see Methods; $t(39) = 9.284$, $p = 2.0e - 11$). This pattern of

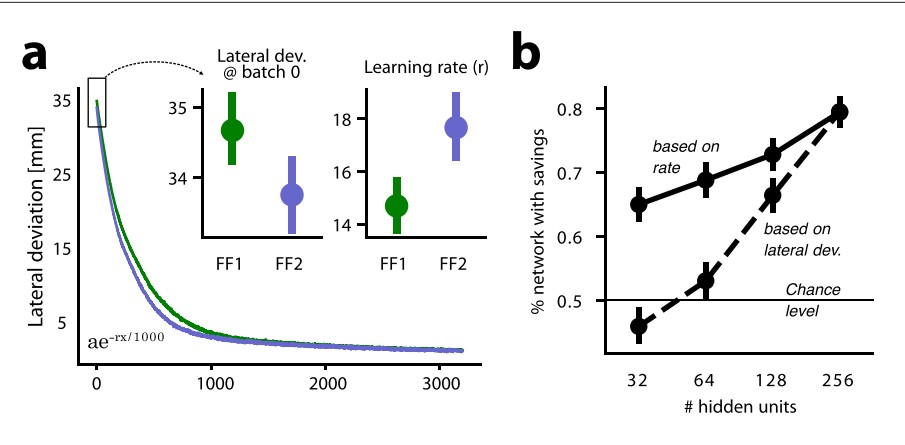

**Figure 3.** Recurrent neural network (RNN) models exhibit behavioral characteristics of savings. (**a**) Mean learning curves averaged over RNN models. Sub-panels show (left) lateral deviation at batch 0 (before training in the corresponding phase) and (right) learning rate *r* after fitting an exponential curve to lateral deviations over training for each network. (**b**) The percentage of networks (n=40 total) with savings is plotted against the number of RNN hidden units. The dashed line indicates the percentage of networks with savings, defined as force field (FF2) learning rate greater than FF1 learning rate. The solid line shows percentage of networks showing savings (lateral deviation at batch 0 smaller in FF2 than FF1). Error bars indicate 95% confidence interval.

improved performance in FF2 is seen in both human and monkey studies of motor learning and is typically referred to as savings.

The difference in performance in the first exposure to FF1 versus the first exposure to FF2 indicates that the network weights changed in such a way that facilitated improved initial performance and faster learning rate in FF2 as compared to FF1, while also not interfering with the performance in NF2. In other words, after training FF1, and washout in NF2, the networks retained enough information about the force field to improve re-learning in FF2, and this retained information was stored in such a way that it did not interfere with the ability of the networks to perform in NF2 just as they had done in NF1, before any FF learning. The pattern of savings observed here in our RNNs occurred despite the absence of any explicit contextual signal indicating the presence or absence of the simulated FF. When contextual signals are present, neural network models often create separate representations for two different tasks (***Driscoll et al., 2022***). We hypothesize that in our study the high dimensionality of the RNNs allows them to develop representations that effectively serve the ongoing task while also preserving some information about previously learned tasks.

We tested this hypothesis by repeating all simulations using RNNs with different numbers of hidden units. We found that as the number of hidden units increases, the likelihood of networks expressing savings also increases. For models with 256 hidden units, there was an 80% chance of expressing savings based on both the learning rate (if the learning rate is faster in FF2 than in FF1) and lateral deviation (if the initial lateral deviation in FF2 is smaller than when FF1 is first encountered). This probability could drop to nearly chance level if the number of hidden units is smaller than 32. This supports the idea that savings may depend upon the dimensionality of the network's weight space.

As an additional control, we trained networks after the growing up phase on an opposing force field (CCW) and then as above, exposed the networks to a NF washout phase, and then to a CW force field. In this case, no savings was observed in the CW force field, either for initial lateral deviation, or for learning rate (***Figure 4***).

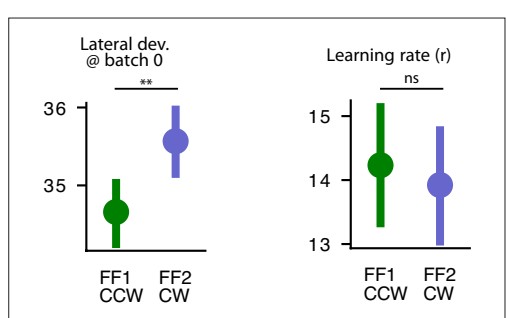

**Figure 4.** No savings observed for novel force field in force field (FF2). (left) The lateral deviation at batch 0 (before training in the corresponding phase) and (right) the learning rate *r* after fitting an exponential curve to the lateral deviations over training for each network.

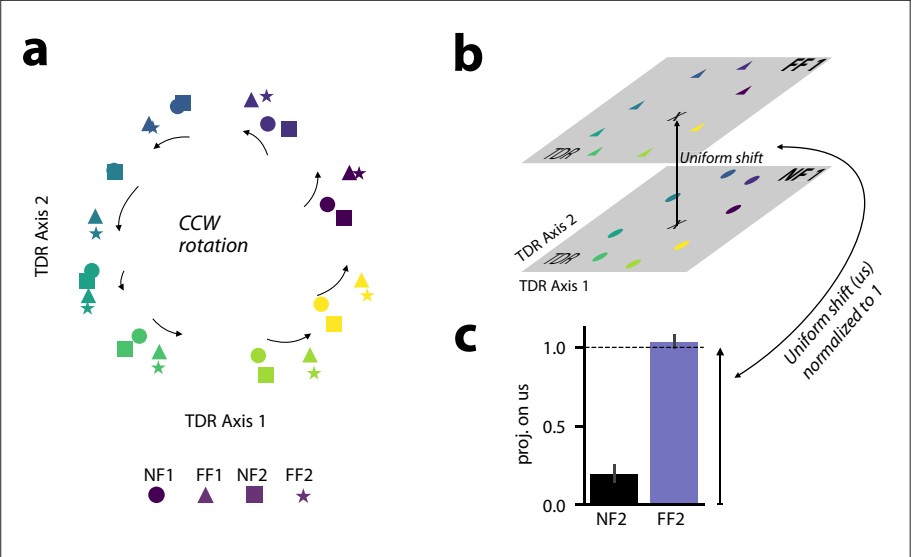

**Figure 5.** Changes in the preparatory activity of recurrent neural network (RNN) hidden units following force field (FF) learning. (**a**) Projection of the hidden preparatory activity (340 ms before the go-cue) of an example trained model performing eight centre-out reaches on the force-predictive subspace acquired with targeted dimensionality reduction (TDR). Different reaching targets are indicated with different colors, and different adaptation phases are indicated with different shapes: circle for null field (NF1), triangle for FF1, square for NF2, and star for FF2. (**b**) A schematic illustration of the uniform shift. Each cross indicates the centre of the hidden preparatory activity for NF1 and FF1, and the arrow indicates the uniform shift. (**c**) Projection of the hidden preparatory activity of all phases onto the uniform shift after orthogonalizing the uniform shift with respect to TDR. The data are scaled so that the projection of NF1 onto the uniform shift is zero and the projection of FF1 is one.

In fact, we observed that initial lateral deviation is larger for the novel force field ($t(39) = -4.918$, $p = 1.6e - 5$). This observation is in line with the finding that learning opposing force fields sequentially results in interference (*Sun et al., 2022*). The results of these control simulations underscore that the savings effect observed in our main study was learning-specific—it was due to prior learning of the CCW force field, and not a general effect of learning any novel dynamics.

## Learning-related changes in hidden unit activity

To characterize changes in hidden unit activity after learning, we followed a similar approach to that described by Sun, O'Shea, and colleagues (*Sun et al., 2022*), in which they investigated how neural population activity in motor cortex changed after monkeys learned to adapt to FFs in an upper limb reaching task. The focus is on hidden unit activity during the preparatory phase, prior to the go signal, as this is the primary determinant of the feed-forward motor commands to muscles (*Churchland and Shenoy, 2024*).

We examined changes in a subspace defined by the relationship between hidden unit activity in the preparatory period, prior to the go cue, and movement-related force at the initial acceleration phase of the movement (hereafter referred to as the TDR subspace, see Methods for further details). We identified the TDR subspace by linearly regressing the preparatory hidden unit activity (340 ms before the go-cue) onto the endpoint (hand) force early during execution (90 ms after the go-cue; see Methods). Projecting the preparatory hidden unit activity associated with all eight centre-out reaches onto this subspace revealed a ring formation (*Figure 5a*). This circular pattern has also been observed in empirical studies of adaptation for motor cortical neurons (*Sun et al., 2022*). This ring rotated in a counter-clockwise (CCW) following adaptation to a clockwise (CW) FF. This is consistent with the idea that after training in the CWFF, the preparatory activity of the RNN hidden units is tuned to facilitate the production of forces in the direction opposite to the upcoming FF during movement.

After washout (NF2), this rotation reverted, leaving little or no residual part of the initial CCW rotation in the preparatory hidden unit activity (*Figure 5a*). The preparatory hidden unit activity again rotated in a CCW direction after adaptation to the CWFF in FF2.

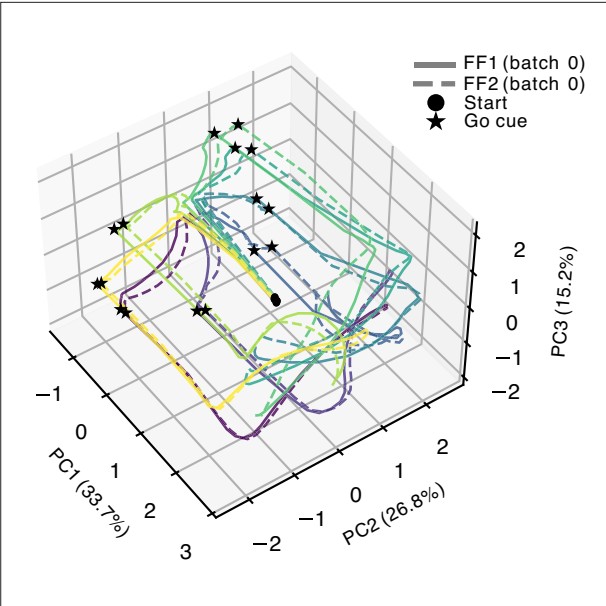

**Figure 6.** Neural trajectories during initial force field (FF1) and FF2. Trajectories for eight movement targets starting at the go cue and ending 200ms into movement execution. Colors show movement directions. PC1–3 represent the first three principal components of neural activity variance.

The neural trajectories for preparation and for movement can be visualized in principal component space. *Figure 6* shows trajectories during planning and early execution for initial FF1 and FF2 exposure. Hidden unit activity was subjected to a principal components analysis, and neural trajectories within the first three PCs are shown for movements to each of the eight movement targets. Filled circles indicate neural state 200 ms prior to the go cue. During the preparatory period trajectories travel along PC1 and then disperse across PC2 and PC3 into the circular pattern indicated by the filled stars, which indicate time of the go cue (also see *Figure 5A*). After the go cue neural trajectories shift back along PC1 and rotate along oscillatory patterns characteristic of populations of motor cortical neurons in non-human primates during movement (*Churchland and Shenoy, 2024*).

We probed the learning-related changes in RNN hidden unit activity that occurred outside of the TDR subspace. To do this, we calculated the extent to which the centroid of the preparatory hidden unit activity for eight centre-out targets shifted following FF learning. Following the procedure used in Sun, O'Shea et al., to isolate learning-related changes in hidden unit activity common to all reach directions we calculated the difference between the mean preparatory activity (340 ms before go cue) of NF1 and FF1 and then orthogonalized this with respect to the TDR subspace (*Sun et al., 2022*). The result is referred to as a 'uniform shift' (*Figure 5b*). After projecting the mean, preparatory activity of all experimental phases onto this uniform shift and normalizing the projection values such that the NF1 projection is 0 and the FF1 projection is 1 (see Methods), we observed that at the end of the washout phase (NF2), the RNN hidden unit activity still showed a projection onto this direction that was significantly different than zero (t(39)=7.484, p=4.6e-9; *Figure 5c*). This indicates that the mean hidden unit activity during the preparatory period did not fully revert to pre-adaptation levels, despite full behavioral washout by the end of the washout phase (*Figure 2a*). This result is consistent with findings in monkey motor cortex (*Sun et al., 2022*), and the idea that this residual hidden unit activity captures information about the previously learned FF, and can be linked to subsequent savings.

## Perturbing preparatory hidden unit activity along the uniform shift

The fact that the preparatory activity of RNN hidden units did not fully revert to the NF1 levels in the uniform shift direction suggests that this residual activity might underlie savings (*Sun et al., 2022*; *Losey et al., 2024*). We tested this idea directly by perturbing hidden unit activity along the direction of the uniform shift, and we probed the effect of these perturbations on behavioral measures of savings.

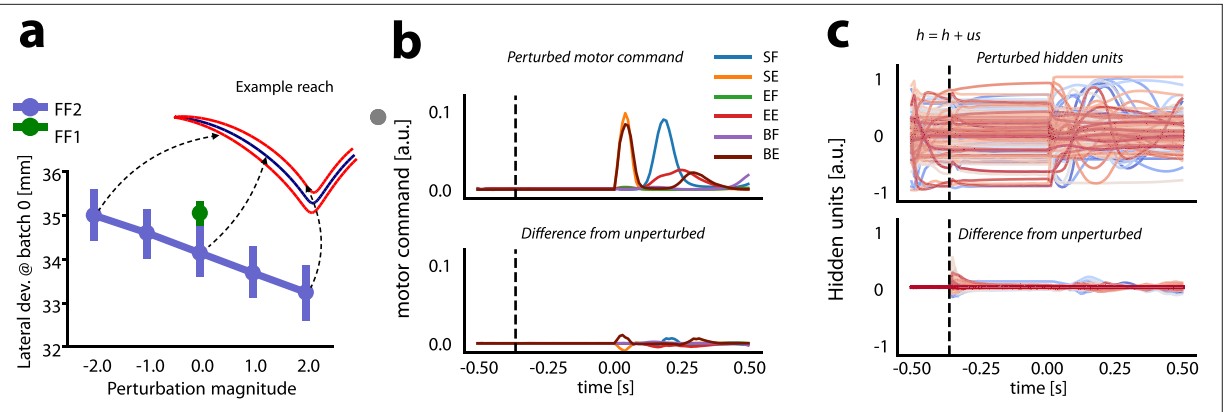

**Figure 7.** Uniform shift in recurrent neural network (RNN) hidden unit activity is related to savings. (**A**) Lateral deviation in force field (FF2) (purple) when hidden preparatory activity was perturbed in the positive (+) and negative (-) directions of the uniform shift with different magnitudes. Arrows indicate trajectories of an example model when hidden unit activity was perturbed (red) or not (blue). Lateral deviation of hand trajectories for FF1 are shown in green. (**B**) Motor commands when hidden unit activity was perturbed with a magnitude of 2.0. Vertical dashed line indicates when the perturbation was delivered. The lower sub-panel shows the difference from the unperturbed motor command. (**C**) Activity of 128 RNN hidden units after perturbation (dashed line indicates time of perturbation along the uniform shift). Lower sub-panel shows the difference between the unperturbed and perturbed RNN hidden unit activity.

We added a proportion of the uniform shift to the preparatory hidden unit activity of networks performing centre-out reaches and we measured the resulting changes in the lateral deviation of simulated hand trajectories. Importantly, the perturbations to preparatory hidden unit activity resulted in little or no changes in muscle activity prior to movement, and during movement itself no perturbations were delivered. We used models at batch 0 of the FF2 phase, when they had not yet been trained on FF adaptation a second time. We have already shown that in FF2, hand trajectory lateral deviation is smaller than in FF1, and we used this as a metric to characterize savings (*Figure 3a*). We examined how much the lateral deviation at batch 0 would change following perturbations to RNN hidden unit activity in the direction of the uniform shift. Details about how the perturbations to hidden unit activity were implemented are found in Methods.

When we perturbed the preparatory hidden unit activity in the opposite direction of the uniform shift (negative magnitudes in *Figure 7a*), lateral deviation of movement-related hand trajectories increased. By increasing the magnitude of perturbation further, we could effectively abolish savings altogether. In contrast, if we perturbed the RNN hidden unit preparatory activity in the same direction as the uniform shift, lateral deviation of hand trajectories was reduced, thus enhancing savings. Example trajectories toward the right-most target are shown in *Figure 7a* for each uniform shift perturbation. These results support the idea that the hidden unit activity in the direction of the uniform shift that remains after washout represents a neural trace of the initial FF learning, one that supports subsequent savings when the models adapt to the FF a second time.

The perturbations changed the activity of hidden units (*Figure 7c*). These changes were large early after the delivery of the perturbation, and then they reached a steady state. However, it is important to note that these changes in the preparatory hidden unit activity did not result in substantive changes in the motor commands (*Figure 7b*), which emphasizes that the uniform shift resides in the null space of motor output.

In summary, the activity of networks along the direction of the uniform shift did not revert fully to pre-adaptation levels after the washout phase, despite the behavior (hand trajectories and muscle activity) being fully washed out, the same as pre-adaptation baseline performance. Perturbing RNN hidden unit activity during the preparatory period along the direction of the uniform shift enhanced savings while perturbations which reduced this residual hidden unit activity reduced or eliminated savings. RNN models with higher dimensionality (more hidden units) were more likely to exhibit evidence of savings while smaller models were less likely to show savings.

# Discussion

We have shown here that RNNs trained to control a computational model of the upper limb show behavioral signatures of savings when learning multiple FFs in succession. We also found that savings is enhanced when the dimensionality of the control network is higher. Following the approach described in a recent electrophysiological study of monkey motor cortex (*Sun et al., 2022*), we identified a subspace of RNN hidden unit activity during the preparatory period after target presentation but prior to movement that shifts after initial FF learning, and subsequently retreats after behavioral washout in a NF. Importantly, this 'uniform shift' did not retreat all the way back to pre-FF baseline levels after washout (see *Figure 5c*). Despite this residual trace of FF learning after washout, in a NF the RNN produced reaches to targets that were the same as those produced prior to initial learning. Importantly, alternating network training on opposing fields (CW and CCW) did not produce savings.

We interpret this residual hidden unit activity as a neural trace of the initial learning that remains after washout, and this is consistent with the idea that this persistent signal contributes to savings (*Sun et al., 2022*). We tested this hypothesis directly by perturbing RNN hidden unit activity during the preparatory period along the direction of the uniform shift—an approach that is possible in a computational model but is not presently feasible in a biological neural network. After NF washout when we re-exposed networks to a previously learned FF, perturbations that amplified the residual trace of learning (increasing activity along the uniform shift) resulted in increased savings. When we perturbed hidden unit activity in the other direction to reduce activity along the uniform shift, savings was reduced and in some cases abolished altogether. These results provide evidence of a causal link between the identified neural trace of learning that remains after washout and subsequent savings when RNNs are re-exposed to the previously learned FF.

Our results are compatible with the proposal by Sun, O'Shea, et al. that the activity of neurons in primary motor cortex during the preparatory period prior to movement contains a component that tracks previously learned motor behavior (*Sun et al., 2022*). They propose that these residual traces of prior learned behavior are encoded in a way that separates the associated motor memories in neural state space and facilitates recall of the appropriate control policies. Losey et al. describe a similar account in the context of a brain-machine interface in which new motor learning is encoded in a neighbouring region of neural state space such that it solves the motor control task but doesn't interfere with prior learning (*Losey et al., 2024*). In our RNNs, this was achieved through learning-related changes in the recurrent weights, such that after NF washout the component of the uniform shift that remained didn't interfere with NF motor behavior, but did produce savings when networks were re-exposed to the previously learned FF.

Our finding that higher-dimensional RNNs are more likely to produce savings supports the idea that encoding newly learned control policies so that they do not interfere with previously learned motor memories depends upon the availability of adequate dimensionality in neural state space. This implies that multiple motor memories can be encoded in neural subspaces so that they do not interfere with each other. A number of recent empirical studies support this idea. Kim et al. recorded from anterior lateral motor cortex of mice over several months and tracked how neurons represented different learned motor tasks (*Kim et al., 2025*). They found that learning produced new neural representations that did not modify existing representations, and re-exposure to a previously learned motor task re-activated the previous neural activity patterns. In a recent paper Bernardi et al. recorded neural activity in prefrontal cortex and hippocampus of monkeys during cognitive tasks, and found that multiple abstract task-related variables were encoded in neural state space using a geometry that allowed separability using linear classifiers (*Bernardi et al., 2020*). Neural recordings in monkey motor cortex show that this kind of task representation emerges prior to movement, in preparatory activity after a movement target is shown but before a go signal instructs the animal to begin a movement (*Churchland et al., 2012*; *Sun et al., 2022*). A similar time course emerged in our RNN simulations here (*Figures 1e and 5a*).

In primates, presumably high-level contextual cues can aid in indexing the appropriate previously learned control policy by activating populations of neurons in a neural direction appropriate for the previously learned task. Indeed a number of theoretical accounts exist that position contextual cues as a driver of motor memory encoding and selection (*Wolpert and Kawato, 1998*; *Haruno et al., 2001*; *Heald et al., 2021*). Similar indexing is likely occurring in accounts where the learning of FFs that would normally interfere is avoided by associating each with the planning (*Sheahan et al., 2016*)

or imagination (*Sheahan et al., 2018*) of different follow-through movements. In our RNNs no such contextual signal was provided, and so the question arises, how are residual traces of previously learned FFs activated? One possibility is that the error signals encountered when our RNNs are re-exposed to a previously learned FF activate neurons within the previously learned subspace. This kind of scheme in which re-exposure to previously encountered errors produces savings is consistent with accounts in which a history of errors or corrections to errors plays a role in motor memory formation (*Herzfeld et al., 2014*; *Leow et al., 2016*).

In a recent computational modeling study, Dubreuil et al. proposed that non-random neural population connectivity structures involving multiple subpopulations that play functionally distinct roles encode multiple tasks better than random connectivity structures (*Dubreuil et al., 2022*). This seems compatible with the idea presented here and in other recent work that low-dimensional recurrent subspaces embedded within a high-dimensional neural control space are used to encode features of movements, such as target directions (*Churchland et al., 2012*), anticipated sensory consequences of perturbations (*Michaels et al., 2024*), and motor skills (*Sun et al., 2022*; *Losey et al., 2024*). The idea that motor memories are encoded in a distributed sensorimotor network and that features of motor adaptation emerge as a result of the dynamical properties of recurrent neural circuits have also been discussed in other computational modelling studies using recurrent neural networks. Ajemian et al. proposed a theoretical framework in which error signals prompt a reorganization of synaptic connectivity to encode motor memories within a high-dimensional neural space (*Ajemian et al., 2010*; *Ajemian et al., 2013*).

The present results are based on RNNs trained in an error-based approach using backpropagation through time (*Werbos, 1990*) using the Adam optimizer (*Kingma and Ba, 2014*). Other techniques for training RNNs have been proposed, including the FORCE algorithm (*Sussillo and Abbott, 2009*). In addition, several recent reports have demonstrated success using reinforcement learning approaches to train neural networks in the context of sensorimotor control tasks (*Lillicrap et al., 2015*; *Codol et al., 2024a*). An interesting avenue for future work is to determine how the present results may or may not generalize to different neural network architectures and learning rules.

The work presented here adds to the growing literature documenting how complex features of motor behavior can arise due to the dynamics of preparatory neural activity in motor cortex. Fuelner et al. show that a number of features of motor adaptation emerge as a result of the dynamical properties of recurrent neural circuits in which sensory feedback modulates motor output (*Feulner et al., 2025*). In a recent paper, Smoulder et al. show that neural activity in monkey motor cortex scales with reward magnitude, and that this reward signal interacts with movement preparation signals in such a way that high rewards disrupt movement preparation and result in poor motor performance compared to moderate rewards (*Smoulder et al., 2024*).

The phenomenon of savings in motor learning implies that motor memory associated with an initial bout of training leads to faster subsequent relearning. The nature of the memory that is stored and how it influences subsequent relearning has been a topic of some debate in the recent literature. One account of savings emphasizes the effect of explicit, strategic components of motor learning (*Huberdeau et al., 2015*; *Morehead et al., 2015*; *Avraham et al., 2021*). Another line of work focuses on the idea that faster relearning may also be driven by implicit learning processes that result from previously experienced errors (*Leow et al., 2016*; *Coltman et al., 2019*; *Yin and Wei, 2020*; *Coltman et al., 2021*).

The simulations described here do not constitute evidence that savings in motor learning tasks is exclusively implicit in animals and humans. The purely context-free learning implemented in our simulations is not meant to be a full model of biological learning, as in biological systems, some form of contextual information is invariably available. Indeed, computational models of motor learning that incorporate contextual effects already exist, e.g. (*Heald et al., 2021*). Nevertheless, our simulations provide a useful window into what the context-free component of savings may look like. This approach offers a powerful means of probing the context-free component of savings in isolation—something that is not readily achievable in animal or human experiments.

Recent empirical work suggests that relearning after washout of implicit adaptation can be attenuated rather than facilitated, a phenomenon attributed to anterograde interference from the washout phase (*Leow et al., 2020*; *Yin and Wei, 2020*; *Hamel et al., 2021*; *Hamel et al., 2022*; *Avraham et al., 2021*; *Hadjiosif et al., 2023*; *Wang and Ivry, 2025*). The savings observed in our simulations

differs from these behavioral findings. Crucially, our model excludes both contextual interference (since no cues signal which force field is present) and explicit-implicit interactions (since context-driven explicit learning is absent). Our goal was not to model a complete explicit-implicit system, but rather to probe how savings may emerge from a purely implicit mechanism and to compare the underlying neural geometry to monkey electrophysiology data. Our results suggest that high-dimensional neural circuits possess an intrinsic capacity for savings via persistent preparatory traces. How and when this capacity may be masked by interference or explicit-implicit interactions in biological systems remains an open question for future work.

The results of our work here with RNNs and the related electrophysiological studies of populations of motor cortical neurons of non-human primates point to a neural basis of savings (**Sun et al., 2022**; **Losey et al., 2024**). We showed here that by increasing the number of hidden units in our RNNs, and hence increasing the dimensionality of the control space, RNNs were more likely to produce behavioral signatures of savings (**Figure 3b**). The high dimensionality of neural space enables a new motor memory to be encoded in such a way that it doesn't interfere with other previously learned information, while still facilitating savings when the network is re-exposed to the newly learned skill. This neural basis of savings can be characterized as implicit, since in our RNN simulations, we did not provide the network with any contextual input that would signal the presence or absence of any given FF.

## Methods

### RNN model

Recurrent neural networks (RNNs) were trained to control movements of a simulated two degree of freedom arm that included rotations about a shoulder joint and an elbow joint in a horizontal plane. The model includes six rigid-tendon Hill-type muscle actuators comprising mono-articular flexors and extensors spanning the shoulder and elbow, as well as a pair of bi-articular muscles producing flexion or extension forces about both shoulder and elbow joints (**Kistemaker et al., 2010**). RNN models are implemented in PyTorch (**Paszke et al., 2019**) and receive a 17-dimensional input signal to a linear input layer, which is fully connected to a recurrent layer consisting of gated recurrent units (GRUs) (**Cho et al., 2014**). The GRU layer is connected to a 6-dimensional linear output layer which provides stimulation commands over time to each of the six muscles in the arm model (see **Figure 1**). Simulations were carried out in Python 3.10 using our open-source MotorNet toolbox (**Codol et al., 2024b**).

Input-hidden ($W_{in}$) and hidden-hidden recurrent ($W_r$) weights (see **Figure 1**) were initialized using Glorot initialization (**Glorot and Bengio, 2010**) and orthogonal initialization (**Hu et al., 2020**), respectively, with biases set to 0. The output layer used a sigmoid activation function. The hidden-output weights ($W_{out}$) were initialized with the Glorot initialization scheme, and its biases were set to –5.0. The sigmoid activation function ensured the controller's output remained close to 0 at the start of training, promoting a stable initialization state. We set the initial hidden unit activity of the network ($\mathbf{h}_0$) as a learnable parameter and initialized it to 0.

The RNN models received a 17-dimensional input vector consisting of task-related inputs along with time-delayed feedback representing visual and proprioceptive signals. The task-related input consisted of a 2-element vector of $(x, y)$ Cartesian coordinates for the target position, and a binary go-cue signal that switched from 0 to 1 when the movement should be initiated. The visual feedback was a Cartesian coordinate of the arm's endpoint $(x, y)$, and the proprioceptive feedback was the lengths and velocities of all six muscles. The time step for simulations was set to 10ms, the visual delay ($\Delta_v$) was 70ms, and the proprioceptive delay ($\Delta_p$) was 20 ms. We also treated the go cue as a visual signal, meaning that at each time step, the network received the 70 ms time delayed value. At each time step, the RNN model transformed the above described 17-dimensional input into a 6-dimensional muscle stimulation command.

### Growing up training phase

During an initial 'growing up' phase new initialized RNN models were trained to move the arm from random starting positions to random target positions, both drawn from a uniform distribution across the joint space of the arm model. Note that due to muscle lengths and joint geometry, only a subset of the workspace was reachable for the model (**Figure 1d**). In 50% of simulations, no go-cue was

provided (a catch trial). This was done to ensure that the network avoided producing anticipatory muscle stimulation commands. In the other 50% of cases, the time of the go-cue switch from 0 to 1 was drawn from a random uniform distribution between 100ms and 300 ms after the start of each simulated trial.

The loss function for training was mainly comprised of position loss, the Euclidean distance between the arm endpoint position $\mathbf{x}_t$ and the desired position $\mathbf{x}_t^*$. The desired position was set to be equal to the starting position of the limb's endpoint before the go cue, and after that the target position. We also included terms in the loss function that punished large and oscillatory hidden and muscle activity, and the jerk (the second derivative of acceleration) of the endpoint trajectory (*Flash and Hogan, 1985*). The full form of the loss function is shown in *Equation 1*:

$$L = \frac{\sum_{t=1}^{N} L_t}{N}$$

$$L_t = 10^3 L_t^p + 10^5 L_t^j + 10^{-1} L_t^m + 10^{-5} L_t^h$$

$$L_t^p = \left\| \mathbf{x}_t^* - \mathbf{x}_t \right\|_1$$

$$L_t^j = \dddot{\mathbf{x}}_t^T \dddot{\mathbf{x}}_t \tag{1}$$

$$L_t^m = \mathbf{f}_t^T \mathbf{f}_t + 3 \times 10^{-3} \dot{\mathbf{f}}_t^T \dot{\mathbf{f}}_t$$

$$L_t^h = \mathbf{h}_t^T \mathbf{h}_t + 10^2 \dot{\mathbf{h}}_t^T \dot{\mathbf{h}}_t$$

where the subscript $t$ indicates time step, $N$ is the total number of time steps in the simulation, $T$ is the transpose operation, and $\|\ \|_1$ is the $L_1$ norm. $L_t^p$, $L_t^j$, $L_t^m$, and $L_t^h$ indicate position, jerk, muscle, and hidden loss, respectively. $\mathbf{h}_t$ is a $n$-element vector of hidden unit activity, and $\dot{\mathbf{h}}_t$ is its derivative. $\mathbf{f}_t$ is a 6-element vector of muscle forces, and $\dot{\mathbf{f}}_t$ is its derivative. Note that muscle forces are different from muscle stimulation commands (*Figure 1A and C*), which are inputs to the Ordinary Differential Equation that produces muscle forces (*Kistemaker et al., 2010*).

The RNN models were initially trained on 20,000 batches with a batch size of 32 on simulations of 1 s (100 time steps). RNN weights were adjusted using the Adam optimization scheme (*Kingma and Ba, 2014*) with a learning rate $lr = 0.003$.

## Motor learning phases

Once the networks were trained to perform reaches to random targets in a null-field (NF), we fixed the input-to-hidden weights ($W_{\text{in}}$) and the hidden-to-output weights ($W_{\text{out}}$) and their biases. This allowed us to isolate subsequent learning-related changes resulting from our experimental manipulations to the recurrent connectivity of hidden units (*Feulner et al., 2025*).

We trained networks on centre-out reaches from a start position to 8 equidistant targets (0.1 m) around the circumference of a circle (see *Figure 1*). The start position corresponded to external joint angles of 60 degrees at the shoulder and 90 degrees at the elbow. We exposed models sequentially to FFs or NFs and in each phase, we continued to adjust hidden recurrent weights to optimize the loss function described in *Equation 1*. During these experimental phases, we used a stochastic gradient descent optimization scheme with learning rate parameter $lr = 0.005$ (*Sutskever et al., 2013*). This ensures batch-local learning, and thus provides greater control and transparency over the course of learning. It also results in gradual learning over batches, better resembling learning curves from empirical studies of force field learning in humans and non-human primates.

In the NF1 experimental phase, the models were trained on centre-out reaches only. We trained the models for 10,000 batches of size 32 (four repetitions of each of the eight targets). As in the growing-up phase, 50% of trials were catch-trials in which the go-cue did not change from 0 to 1. After NF1 training, we continued to train the models to perform centre-out reaches but we introduced a clockwise curl force field (CWFF) for all movements (FF1). The external force ($F_x$, $F_y$) applied at the arm's endpoint that produced a CWFF is described by the following equation:

$$\begin{bmatrix} F_x \\ F_y \end{bmatrix} = b \begin{bmatrix} 0 & -1 \\ 1 & 0 \end{bmatrix} \begin{bmatrix} \dot{x} \\ \dot{y} \end{bmatrix} \tag{2}$$

where $\dot{x}$ and $\dot{y}$ are the velocity of the arm's endpoint in Cartesian coordinates and $b = 8$ Ns/m is a scalar constant defining the strength of the FF. In the null field (NF), $b = 0$. We trained the models for 3200 batches of size 32 in the FF1 experimental phase, with 50% catch-trials.

Following FF1, the models were trained again in a null field (NF2), using the same procedures as in NF1 (10,000 batches of size 32). After NF2, the models were again trained in the presence of a CWFF (FF2), exactly as in FF1.

## Lateral deviation

We evaluated the behavioral performance of the models during NF1, FF1, NF2, and FF2 by calculating the maximum lateral deviation of the endpoint trajectory from straight lines connecting the starting and target positions. We will refer to this measure as 'lateral deviation,' and it was considered positive if it was in the clockwise direction from the straight line, and negative otherwise. For each batch of training, we calculated the mean lateral deviations across all eight targets.

For each model, we characterized the learning rate during FF1 and FF2 by fitting an exponential function of the following form to the mean lateral deviation across training batches:

$$\hat{y} = \alpha e^{-rx_n/1000} \tag{3}$$

where $\hat{y}$ is the modelled lateral deviation at batch number $x_n$, $\alpha$ is a scaling factor that determines the initial lateral deviation, and $r$ is the rate at which lateral deviation decays over time (indicating the learning rate). Before fitting we smoothed learning rates over batches by window-averaging with a kernel size of five batches.

## Targeted dimensionality reduction

Following the procedure described in *Sun et al., 2022*, we identified a subspace of RNN hidden unit activity that predicts the arm's endpoint initial forces based on the preparatory activity of hidden units (hidden unit activity before the go-cue). To do this, we applied targeted dimensionality reduction (TDR) using model data at the end of the NF1 experimental phase. The subspace is defined as:

$$\mathbf{H}_{-340\,\text{ms}}^{\text{NF1}} = \begin{bmatrix} \mathbf{F}_{+90\,\text{ms}}^{\text{NF1}} & \mathbb{1} \end{bmatrix} W \tag{4}$$

where $\mathbf{H}_{-340\,\text{ms}}^{\text{NF1}}$ is the matrix of hidden unit activity of size (targets × units) 340 ms before the go-cue, $\mathbf{F}_{+90\,\text{ms}}^{\text{NF1}}$ is the matrix of endpoint forces of size (targets × 2) 90 ms after go-cue, $\mathbb{1}$ is the targets-element vector concatenated to the force matrix, and $W$ is the matrix of size (3 × units). For the $\mathbf{F}_{+90\,\text{ms}}^{\text{NF1}}$ parameter, 90 ms was chosen because it coincides with peak acceleration. For the $\mathbf{H}_{-340\,\text{ms}}^{\text{NF1}}$ parameter, 340 ms was chosen because it ensures that hidden unit activity is stabilized.

We then calculated the pseudo-inverse of $W$, resulting in a (units × 3) matrix $W^+$. To find the force-predictive subspace, we took the first two columns of $W^+$ (ignoring the intercept) and orthogonalized them using the Gram-Schmidt orthogonalization scheme, resulting in $\hat{W}^+$. We projected the preparatory hidden unit activity of all experimental phases ($\mathbf{H}_{-340\,\text{ms}}^{\text{NF1}}, \mathbf{H}_{-340\,\text{ms}}^{\text{FF1}}, \mathbf{H}_{-340\,\text{ms}}^{\text{NF2}}, \mathbf{H}_{-340\,\text{ms}}^{\text{FF2}}$) onto $\hat{W}^+$ after removing their global mean.

## Uniform shift

Following the experimental phases FF1 and FF2, we calculated the direction in which the RNN hidden unit activity during the preparatory period shifted, averaged across movement directions. We averaged the preparatory hidden unit activity over the 8 targets in FF1 and NF1, and then calculated the difference. Following the convention in *Sun et al., 2022*, this shift is referred to as a 'uniform shift' (us):

$$\mathbf{us} = \bar{\mathbf{H}}_{-340\,\text{ms}}^{\text{FF1}} - \bar{\mathbf{H}}_{-340\,\text{ms}}^{\text{NF1}}, \tag{5}$$

where the bar indicates averaging over 8 movement directions. We orthogonalized the uniform shift with respect to $\hat{W}^+$. This allowed us to test for changes outside the force-predictive subspace.

We then projected the preparatory hidden unit activity (340 ms before the go-cue) of all experimental phases after removing the global mean. Next, we normalized the projection values for each

model so that the projection of $\mathbf{H}^{NF1}$-340 ms onto the uniform shift is zero, and the projection of $\mathbf{H}^{FF1}$-340 ms onto the uniform shift is one.

### Perturbing hidden unit activity

To conduct a causal test of the idea that the non-zero uniform shift activity that remained following NF2 is related to savings, we perturbed the activity of hidden units at the end of the preparatory period by adding to each hidden unit a proportion ($-2$, $-1$, $0$, $1$, $2$) of the projection of that unit onto the uniform shift. We conducted these perturbations separately for each movement direction. The perturbation was applied 340ms prior to the go cue and the duration of the perturbation was one simulation time step.

## Acknowledgements

This work was supported by the Natural Sciences and Engineering Research Council of Canada through a Discovery Grant RGPIN/05458-2018 to PLG, and a FRQNT Strategic Clusters Program grant to OC JAM was supported by a Banting Postdoctoral Fellowship and a BrainsCAN Postdoc toral Fellowship, and by the Canadian Institutes of Health Research grant PJT-175010 to Dr. Andrew Pruszynski. The authors wish to thank Mehrdad Kashefi for useful discussions about this project.

## Additional information

### Funding

| Funder | Grant reference number | Author |
|---|---|---|
| Natural Sciences and Engineering Research Council of Canada | RGPIN/05458-2018 | Paul L Gribble |
| Fonds de recherche du Québec | | Olivier Codol |
| Banting Research Foundation | | Jonathan A Michaels |
| Canada First Research Excellence Fund | | Jonathan A Michaels |

The funders had no role in study design, data collection and interpretation, or the decision to submit the work for publication.

### Author contributions

Mahdiyar Shahbazi, Conceptualization, Resources, Data curation, Software, Formal analysis, Validation, Investigation, Visualization, Methodology, Writing – original draft, Writing – review and editing; Olivier Codol, Conceptualization, Resources, Software, Formal analysis, Validation, Investigation, Visualization, Methodology, Writing – review and editing; Jonathan A Michaels, Paul L Gribble, Conceptualization, Resources, Data curation, Software, Formal analysis, Supervision, Funding acquisition, Validation, Investigation, Visualization, Methodology, Writing – original draft, Project administration, Writing – review and editing

### Author ORCIDs

Mahdiyar Shahbazi ⬥ https://orcid.org/0000-0002-4883-4376
Olivier Codol ⬥ https://orcid.org/0000-0003-0796-5457
Jonathan A Michaels ⬥ https://orcid.org/0000-0002-5179-3181
Paul L Gribble ⬥ https://orcid.org/0000-0002-1368-032X

Reviewer #2 (Public review): https://doi.org/10.7554/eLife.107423.3.sa1
Author response https://doi.org/10.7554/eLife.107423.3.sa2

## Additional files

### Supplementary files
MDAR checklist

### Data availability
Python code to reproduce the simulations and analyses described here is available on GitHub at the following repository: https://github.com/mshahbazi1997/MotorSavingModel (copy archived at *Shahbazi, 2026*).

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
