## [Editor Report · eLife Assessment]

This study presents **valuable** computational findings on the neural basis of learning new motor memories and the savings using recurrent neural networks. The evidence supporting the claims of the authors is **solid**, but it would benefit from more detailed discussion on the specific conditions under which savings emerges from purely implicit mechanisms. This work will be of interest to computational and experimental neuroscientists working in motor learning.

---

## [Referee Report · Reviewer #2 (Public review)]

Summary:

Shahbazi et al. trained recurrent neural networks (RNNs) to simulate human upper limb movement during adaptation to a force field perturbation. They demonstrated that throughout adaptation, the pattern of motor commands to the muscles of the simulated arm changed, allowing the perturbed movements to regain their typical, perturbation-free straight-line paths. After this initial learning block (FF1), the network encountered null-fields to wash out the adaptation, before re-experiencing the force in a second learning block (FF2). Upon re-exposure, the network learned faster than during initial learning, consistent with the savings observed in behavioral studies of adaptation. They also found that as the number of hidden units in the RNN increased, so did the probability of exhibiting savings. The authors concluded that these results propose a neural basis for savings that is independent of context and strategic processes.

Strengths:

The paper addresses an important and controversial topic in motor adaptation: the mechanism underlying motor memory. The RNN simulation reproduces behavioral hallmarks of adaptation, and it provides a useful illustration of the pattern of muscle activity underlying human-like movements under both normal and perturbing conditions. While the savings effect produced by the network, though significant, appears somewhat small, the simulation demonstrating an increase in savings with a greater number of hidden units is particularly intriguing.

Main weakness:

The introduction details the ongoing debate in the literature regarding the mechanisms underlying savings, particularly whether it stems from explicit or implicit learning processes. However, it remains unclear how the current work addresses this debate. There is already a considerable body of research, particularly in visuomotor adaptation, demonstrating that savings is predominantly driven by explicit strategies (e.g., Morehead et al. 2015, Haith et al., 2015; Huberdeau et al., 2019; Avraham et al., 2021). Furthermore, there have been multiple reports that implicit adaptation exhibits attenuation upon relearning (Avraham et al., 2021, Leow et al., 2020; Yin and Wei, 2020; Hamel et al., 2021; Hamel et al., 2022; Wang and Ivry, 2023; Hadjiosif et al., 2023). In the discussion, the authors acknowledge that their goal was not to model a complete explicit-implicit system, but rather to probe how savings may emerge from a purely implicit mechanism. Given the central debate introduced by the authors, the manuscript would benefit from a more detailed discussion explaining how their findings elucidate the specific conditions under which savings emerge from purely implicit mechanisms versus when cognitive strategies predominate.

---

## [Author Response]

The following is the authors’ response to the original reviews.

**Public Reviews:**

**Reviewer #1 (Public review):**
Summary:Shahbazi et al used a recurrent neural network model trained to control a musculoskeletal model of the arm to investigate how neural populations accommodate activity patterns underpinning savings. The paper draws upon the recent finding of a "uniform shift" in preparatory activity in monkey motor cortex associated with savings, and leverages full access to a computational model to establish causality.Strengths:The paper is well written, and the figures are clearly presented. The key finding that the uniform shift first reported based on neural recordings by Sun et al. emerges in artificial neural networks performing a similar task is interesting and well-backed by their analyses. Manipulating this uniform shift to show that it drives behavioural savings is an important causal confirmation of the proposal by Sun et al.Weaknesses / Comments:As mentioned earlier, the core results are well backed by the analyses. Most of my comments relate to adding more controls and additional questions that could be explored with the model to strengthen the paper.(1) Savings are quantified as more rapid relearning of the FF upon re-exposure (e.g., Figure 3). This finding is based on backpropagation through time, but would this hold when using a different optimiser, e.g., FORCE?

This is an interesting question, and indeed, there are an increasing number of studies addressing how different neural network learning rules may affect the kinds of representations that arise after learning (Codol et al., 2024). However the focus of the present paper is not on which neural network approaches or which specific optimisers produce savings, rather, the focus is on the basis and neural geometry of savings when it emerges.

We have added a short paragraph to the Discussion section [lines 349-355] to address this:

“The present results are based on RNNs trained in an error-based approach using backpropagation through time (Werbos, 1990) using the Adam optimizer (Kingma and Ba, 2014). Other techniques for training RNNs have been proposed including the FORCE algorithm (Sussillo and Abbott, 2009). In addition, several recent reports have demonstrated success using reinforcement learning approaches to train neural networks in the context of sensorimotor control tasks (Lillicrap et al., 2015; Codol et al., 2024a). An interesting avenue for future work is to determine how the present results may or may not generalize to different neural network architectures and learning rules.”

(2) The authors should include a "null model" showing that training on a different reaching task following NF, as opposed to FF2, won't show something akin to a uniform shift during preparation due to the adoption of TDR and having similar targets.

This is a critical point. Training on a different reaching task other than FF2 (e.g. a different force field) will indeed result in a uniform shift, but critically, a shift in a different direction in neural state space than the uniform shift associated with FF2. The central focus of the present paper is to show that when there remains a non-zero projection of preparatory neural activity along the direction of the uniform shift associated with a given learning task, this residual projection underlies savings when networks are subsequently re-exposed to the same task.

In the Results section we had included a short paragraph to describe control simulations that we performed that address this concept. We have expanded this text and added a Figure and the results of statistical tests to better describe this control [lines 179-187]:

“As an additional control we trained networks after the growing up phase on an opposing force field (CCW) and then as above, exposed the networks to a NF washout phase, and then to a CW force field. In this case no savings was observed in the CW force field, either for initial lateral deviation, or for learning rate (Figure 3). In fact, we observed that initial lateral deviation is larger for the novel force field (t(39)=-4.918, p=1.6e-5). This observation is in line with the finding that learning opposing force fields sequentially results in interference (Sun et al., 2022). The results of these control simulations underscore that the savings effect observed in our main study was learning-specific—it was due to prior learning of the CCW force field, and not a general effect of learning any novel dynamics.”

(3) The analyses of network activity during movement preparation (Figure 4) nicely replicate the key finding in Sun et al, but I think the authors could leverage the full access to their network and go further, e.g., by examining changes (or the lack of) during execution in FF2 with respect to FF (and perhaps in a future NF2 with respect to NF), including whether execution activity lives also lives in parallel hyperplanes, etc.

We agree that a visualization of the neural activity during movement would be beneficial to the reader. To address this we have added a new Figure (Fig. 6) and associated text [lines 210-219]. The Figure shows the neural trajectories when the RNNs are first exposed to the FF1 and when they are first exposed to FF2 (after NF2 washout). Trajectories are plotted in 3D corresponding to the first 3 principal components, starting at the go cue and ending 200 ms into the movement, for each of the 8 movement targets.

“The neural trajectories for preparation and for movement can be visualized in principal component space. Figure 6 shows trajectories during planning and early execution for initial FF1 and FF2 exposure. Hidden unit activity was subjected to a principal components analysis, and neural trajectories within the first three PCs are shown for movements to each of the eight movement targets. Filled circles indicate neural state 200 ms prior to the go cue. During the preparatory period trajectories travel along PC1 and then disperse across PC2 and PC3 into the circular pattern indicated by the filled stars, which indicate time of the go cue (also see Figure 5A). After the go cue neural trajectories shift back along PC1 and rotate along oscillatory patterns characteristic of populations of motor cortical neurons in non-human primates during movement (Churchland and Shenoy, 2024).”

(4) Related to the above, while the results are interesting and the paper is well done, I kept wishing that the authors had done "more" with their model. This could be one or two final sections on "predictions" that would nicely complement their "validation" of the uniform shift, and that, in my opinion, would greatly increase the impact of the paper. In particular:(a) What would be the effect of learning more "tasks"? For example, is there a limit on how many fields can be learned? (You show something related by manipulating network size, but this is slightly different.)

These are interesting questions and to some extent they are already addressed in the paper. Of course, the number of tasks that a network is able to learn, will be related to how much those tasks overlap in a control space. Indeed, this idea goes back to early theoretical accounts of connectionist models such as Hopfield nets and capacity for representing information (Hopfield, 1982; Hopfield et al., 1983). The control simulations that we described in the paper [lines 179-187 and Figure 4] are a test of one extreme version of this, in which two tasks are in direct opposition to each other (opposite force fields), and in this situation no savings emerges. We believe it is an interesting question, but beyond the scope of the present paper to undertake a comprehensive exploration of the nature of task-overlap in upper limb reaching learning tasks.

(b) Figure 5 is a nice causal demonstration that the uniform shift is related to savings. However, and related to comment #3, it'd be interesting to see more details about how the behaviour and the network activity changes as preparatory activity shifts along this axis, in particular regarding how moving the preparatory states affect the organisation and dynamics of upcoming execution activity -these are the kind of intuitions that modelling studies like this one can provide.

This has been addressed above by the changes we made to address the reviewer’s comment #3.

(c) The authors focus on a task design that spans baseline, FF, NF, FF2 to replicate the original study by Sun et al. However, it would be interesting if they generated predictions for neural changes to other types of tasks that have been studied behaviourally. These could include, for example: (i) modelling a visuomotor rotation or a mirror reversal task; (ii) having to adapt to a FF in the opposite direction; (iii) investigating the role of adding an explicit context and having the networks learn multiple FF; and (iv) trying to learn FF fields in opposite directions, perhaps restricted to specific targets. As the authors know, all these questions and more have been studied with similar behavioural paradigms, and it would be nice to see what neural predictions are generated by this model.

See responses above e.g. to comment 4. We have clarified the text and provided a new Figure to illustrate our opposite FF control simulations. The other suggestions about visumotor rotations, and contextual cues, are interesting and potentially important questions that we are working on, but we believe are beyond the scope of the current paper which is focused specifically around the question of savings in FF learning.

(5) On the Discussion: When extrapolating from neural network results to animals, the fact that your networks can learn implicitly doesn't mean that animals do learn implicitly. Indeed, I think the consensus view is that different perturbations may lead to the expression of different types of savings (e.g., FF vs VR, which seems to be more explicit). Besides, these different mechanisms may be primarily implemented by brain regions less directly tied to motor control (e.g., cerebellum, parietal cortex?), which are not directly implemented in the authors' model.

Of course the reviewer is correct that our simulations are not evidence that savings in motor tasks learned by animals is only implicit, and we do not make any such claims in the paper. The model we describe in the present paper is not meant to be a comprehensive model of motor learning in humans/animals. Indeed, the pure “context free” type of learning that we implement in our simulations basically cannot occur in animals, because there is always some information that provides contextual information. Indeed there are computational models of motor learning that include these effects, e.g. the COIN model (Heald et al., 2021). Our model however provides a useful window into what the context-free component of savings may look like. The approach we describe in the present paper is a powerful way to probe the context-free component of savings in isolation in a way that is not possible (at least not readily) in animals/humans. We have modified the text in the Discussion [lines 372-379] to better articulate this point.

“The simulations described here do not constitute evidence that savings in motor learning tasks is exclusively implicit in animals and humans. The purely context-free learning implemented in our simulations is highly unrealistic, as some form of contextual information is invariably available. Indeed, computational models of motor learning that incorporate contextual effects already exist, e.g. (Heald et al. 2021). Nevertheless, our simulations provide a useful window into what the context-free component of savings may look like. This approach offers a powerful means of probing the context-free component of savings in isolation—something that is not readily achievable in animal or human experiments.”

**Reviewer #2 (Public review):**
Summary:Shahbazi et al. trained recurrent neural networks (RNNs) to simulate human upper limb movement during adaptation to a force field perturbation. They demonstrated that throughout adaptation, the pattern of motor commands to the muscles of the simulated arm changed, allowing the perturbed movements to regain their typical, perturbation-free straight-line paths. After this initial learning block (FF1), the network encountered null-fields to wash out the adaptation, before re-experiencing the force in a second learning block (FF2). Upon re-exposure, the network learned faster than during initial learning, consistent with the savings observed in behavioral studies of adaptation. They also found that as the number of hidden units in the RNN increased, so did the probability of exhibiting savings. The authors concluded that these results propose a neural basis for savings that is independent of context and strategic processes.Strengths:The paper addresses an important and controversial topic in motor adaptation: the mechanism underlying motor memory. The RNN simulation reproduces behavioral hallmarks of adaptation, and it provides a useful illustration of the pattern of muscle activity underlying human-like movements under both normal and perturbing conditions. While the savings effect produced by the network, though significant, appears somewhat small, the simulation demonstrating an increase in savings with a greater number of hidden units is particularly intriguing.Weaknesses:(1) To be transparent, savings in motor adaptation have been a primary focus of my own research. Some core findings presented in this paper are at odds with the ideas I and others have previously put forward. While I don't want to impose my agenda on the authors of this paper, I do think the authors should address these issues.(a) The authors acknowledge the ongoing debate in the literature regarding the mechanisms underlying savings, particularly whether it stems from explicit or implicit learning processes. However, it remains unclear how the current work addresses this debate. There is already a considerable body of research, particularly in visuomotor adaptation, demonstrating that savings is predominantly driven by explicit strategies. For example, when people are asked to report their strategy, they recall a strategy that was useful during the first learning block (Morehead et al. 2015). Furthermore, savings are abolished under experimental manipulations designed to eliminate strategic contributions (e.g., Haith et al., 2015; Huberdeau et al., 2019; Avraham et al., 2021). The authors briefly state that their findings support the hypothesis that a neural basis of memory retention underlying savings can be independent of cognitive or strategic learning components, and that savings can be characterized as implicit. While these statements may be true, it is not clear how this work substantiates these claims.

We have addressed a similar point raised by Reviewer 1, see point #5 above. Our work represents an example of how savings can occur from implicit mechanisms in the absence of explicit contextual cues. Our goal is not to resolve the debate about how this occurs in humans/animals. Rather, our model provides a useful window into what the context-free component of savings may look like. Our approach is a powerful way to probe the context-free component of savings in isolation in a way that is not possible (at least not readily) in animals/humans. We have modified the text in the Discussion [lines 372-379] to better articulate this point.

“The simulations described here do not constitute evidence that savings in motor learning tasks is exclusively implicit in animals and humans. The purely context-free learning implemented in our simulations is not meant to be a full model of biological learning, as in biological systems some form of contextual information is invariably available. Indeed, computational models of motor learning that incorporate contextual effects already exist, e.g. (Heald et al. 2021). Nevertheless, our simulations provide a useful window into what the context-free component of savings may look like. This approach offers a powerful means of probing the context-free component of savings in isolation—something that is not readily achievable in animal or human experiments.”

(b) Our research has also demonstrated that if implicit adaptation is completely washed out after the initial learning block, it not only fails to exhibit savings but is actually attenuated relative to the first learning block (Avraham et al., 2021). This phenomenon of attenuation upon relearning can also be seen in other studies of visuomotor adaptation (e.g., Leow et al., 2020; Yin and Wei, 2020; Hamel et al., 2021; Hamel et al., 2022; Wang and Ivry, 2023; Hadjiosif et al., 2023). More recently, we have shown that this attenuation is due to anterograde interference arising from the experience with the washout block experience (Avraham and Ivry, 2025). We illustrated that the implicit system is highly susceptible to interference; it doesn't require exposure to salient opposite errors and can occur even following prolonged exposure to veridical feedback. The central thesis of this paper, namely that implicit savings can emerge through RNNs, is at odds with these empirical results. The authors should address this discrepancy.

These empirical results are interesting and intriguing, and we agree that they are relevant in the context of the debate about the relative contributions and interactions between explicit and implicit learning systems and savings. Importantly, contextual interference is impossible in our model, since there are no contextual cues about which force field is present or absent. Interactions between an explicit system and an implicit learning system are also impossible in our model, since there is no possibility of context-driven explicit learning or memory. The approach we have taken in the present paper is not to model a full explicit plus implicit learning system but rather to probe how savings may emerge from a purely implicit learning mechanism alone and to compare the neural geometry underlying this implicit-drive savings to the neural recording results from monkey electrophysiology studies. Nevertheless we have added some text to the Discussion [lines 380-391] to situate our findings in the context of the studies mentioned above by the reviewer.

“Recent empirical work suggests that relearning after washout of implicit adaptation can be attenuated rather than facilitated, a phenomenon attributed to anterograde interference from the washout phase (Avraham et al., 2021; Hadjiosif et al., 2023; Hamel et al., 2022, 2021; Leow et al., 2020; Wang and Ivry, 2025; Yin and Wei, 2020). The savings observed in our simulations differs from these behavioral findings. Crucially, our model excludes both contextual interference (since no cues signal which force field is present) and explicit-implicit interactions (since context-driven explicit learning is absent). Our goal was not to model a complete explicit-implicit system, but rather to probe how savings may emerge from a purely implicit mechanism and to compare the underlying neural geometry to monkey electrophysiology data. Our results suggest that high-dimensional neural circuits possess an intrinsic capacity for savings via persistent preparatory traces. How and when this capacity may be masked by interference or explicit-implicit interactions in biological systems remains an open question for future work.”

(2) This brings me to the question about neural correlates: The results are linked to activity in the primary motor cortex. How does that align with the well-established role of the cerebellum in implicit motor adaptation? And with the studies showing that savings are due to explicit strategies, which are generally associated with prefrontal regions?

The modeling approach we use in the present paper is area agnostic, and we do not include different neural modules to represent specific brain areas such as cerebellum or prefrontal regions. In the current approach we specifically exclude explicit strategies, as a way to specifically probe implicit mechanisms alone. Also see response to reviewer 1 comment 5 above.

(3) The analysis on the complexity of the neural network (i.e., the number of hidden units) and its relationship to savings is very interesting. It makes sense to me that more complex networks would show more savings. I'm not sure I follow the author's explanation, but my understanding is that increased network complexity makes it more difficult to override the formed memory through interference (e.g., from the experience with NF2). Also, the results indicate that a network with 32 units led to a less-than-chance level of networks exhibiting savings (Figure 3b). What behavioral output does this configuration produce? Could this behavior manifest as attenuation upon relearning? Furthermore, if one were to examine an even smaller, simpler network (perhaps one more closely reflecting cerebellar circuits), would such a model predict attenuation rather than savings?

These are interesting questions, and are potentially important, for future work to explore. Our interpretation of the results of smaller networks is that these small RNNs fail to show savings presumably because the learned FF behavior is 'erased' during washout because of the limited capacity to retain the FF learning in a distinct neighborhood in neural state space. Our paper is focused specifically on the relationship between savings, implicit learning, and neural capacity via network size, in the context of the monkey electrophysiology results in motor cortex. It would be interesting in future work to explore a cerebellar-like modeling approach.

(4) The authors emphasize that their network did not receive any explicit contextual signals related to the presence or absence of the force field (FF), thus operating in a 'context-free' manner. From my understanding, some existing models of context's role in motor memories (e.g., Oh and Schweighofer, 2019; Heald et al., 2021) propose that memory-related changes can be observed even without explicit contextual information, as contextual changes can be inferred from sudden or significant environmental shifts (e.g., the introduction or removal of perturbations). Given this, could the observed savings in the current simulation be explained by some form of contextual retrieval, inferred by the network from the re-presentation of the perturbation in FF2?

It is important to note that this is not possible in the context of the modeling approach described in the present paper. For example, in trial 1 of FF2, because the network has no contextual cue signaling the FF’s presence, the network has no information before movement begins that a FF will be present during movement (recall that the FF is velocity-dependent, and so is zero before movement begins). Once the network encounters the FF during movement, some component of its response I suppose could be described as contextual inference derived from effector state (similar to the account described in the COIN model), but strictly speaking the model is only responding to what it encounters in the moment. Any change in behaviour due to prior learning (e.g. savings) is due to the interaction between the residual learning-related neural state (e.g. the uniform shift), the effector state in the moment, and the errors encountered during movement. We don’t interpret this as “inference” in the traditional sense of an explicit learning system.

(5) If there is residual hidden unit activity related to the FF at the end of the NF2 phase, how does the simulated movement revert back to baseline? Are there any differences in the movement trajectory, beyond just lateral deviation, between NF1 and NF2? The authors state that "changes in the preparatory hidden unit activity did not result in substantive changes in the motor commands (Figure 5b), which emphasizes that the uniform shift resides in the null space of motor output." However, Figure 5b appears to show visible changes in hidden unit activity. Don't these changes reflect a pattern of muscle activity that is the basis for behavior? These changes are indeed small, but it seems that so is the effect size for savings (Figure 3a). Could this suggest that there is not, in fact, a complete washout of initial learning during NF2 within the network?

This is precisely the point of the paper, i.e. to show that neural activity during the preparatory period before movement onset is different, even though the behaviour during the preparatory period is the same (i.e. no muscle activity and no movement). This recapitulates the empirical findings from the neural data reported in the Sun et al. (2022) paper.

The reviewer asks “Don't these changes reflect a pattern of muscle activity that is the basis for behavior?” Yes indeed they do, but not during the NF and not during the preparatory activity prior to movement onset.

The reviewer asks “Could this suggest that there is not, in fact, a complete washout of initial learning during NF2 within the network?” We addressed this in the paper (Results/Washout) by comparing kinematics after washout to that prior to FF learning; e.g. any differences in lateral deviation of the hand path for the entire reach trajectory was in the range of 0.1 mm, which is less than 0.25 % of the lateral deviation encountered in the FF and only 0.1 % of the reach distance (10 cm).

**Recommendations for the authors:**

**Reviewer #2 (Recommendations for the authors):**
(1) Figure 1c, lower panel: Is this from the early or late stage of FF1?

This is an example movement after learning in a null field (NF). We have clarified this in the Figure caption.

(2) Please clarify what the two panels in Figure 1e represent.

We have clarified in the Figure caption that these are activity from two example hidden units.

(3) If Figure 2c is intended to illustrate the changes in motor commands for individual muscles, consider reorganizing the plots by muscle to more clearly show the change for each muscle from NF1 to FF1.

The point here is not to make fine-grained comparisons between specific muscles, rather to show a general example of how muscle activity is different. For the sake of visual simplicity in a Figure that already has many components we have decided to keep Figure 2c the same.

(4) The text mentions that no savings were observed when the network was trained on CCW followed by CW perturbations. However, no data or statistical analysis is presented to support this claim. I wonder if the authors would expect attenuated learning when exposed to the CW perturbation, given a memory of the opposite perturbation.

We have added a Figure to provide data for the FF opposite control.

(5) The relevance of the discussion on choking under pressure to the paper wasn't clear.

We have modified the relevant text in the Discussion section [lines 356-363] to clarify the relevance of the present work to other recent work on how complex features of motor behaviour can arise due to the dynamics of preparatory neural activity in motor cortex.

References

Avraham G, Morehead JR, Kim HE, Ivry RB. 2021. Reexposure to a sensorimotor perturbation produces opposite effects on explicit and implicit learning processes. PLoS Biol 19:e3001147. doi:10.1371/journal.pbio.3001147

Codol O, Krishna NH, Lajoie G, Perich MG. 2024. Brain-like neural dynamics for behavioral control develop through reinforcement learning. bioRxiv. doi:10.1101/2024.10.04.616712

Hadjiosif AM, Morehead JR, Smith MA. 2023. A double dissociation between savings and long-term memory in motor learning. PLoS Biol 21:e3001799. doi:10.1371/journal.pbio.3001799

Hamel R, Dallaire-Jean L, De La Fontaine É, Lepage JF, Bernier PM. 2021. Learning the same motor task twice impairs its retention in a time- and dose-dependent manner. Proc Biol Sci 288:20202556. doi:10.1098/rspb.2020.2556

Hamel R, Lepage J-F, Bernier P-M. 2022. Anterograde interference emerges along a gradient as a function of task similarity: A behavioural study. Eur J Neurosci 55:49–66. doi:10.1111/ejn.15561

Heald JB, Lengyel M, Wolpert DM. 2021. Contextual inference underlies the learning of sensorimotor repertoires. Nature 600:489–493. doi:10.1038/s41586-021-04129-3

Hopfield JJ. 1982. Neural networks and physical systems with emergent collective computational abilities. Proc Natl Acad Sci U S A 79:2554–2558. doi:10.1073/pnas.79.8.2554

Hopfield JJ, Feinstein DI, Palmer RG. 1983. “Unlearning” has a stabilizing effect in collective memories. Nature 304:158–159. doi:10.1038/304158a0

Leow L-A, Marinovic W, de Rugy A, Carroll TJ. 2020. Task errors drive memories that improve sensorimotor adaptation. J Neurosci 40:3075–3088. doi:10.1523/JNEUROSCI.1506-19.2020

Wang T, Ivry RB. 2025. Contextual effects during sensorimotor adaptation are an emergent property of population coding in a cerebellar-inspired model. Sci Adv 11:eadr4540. doi:10.1126/sciadv.adr4540

Yin C, Wei K. 2020. Savings in sensorimotor adaptation without an explicit strategy. J Neurophysiol 123:1180–1192. doi:10.1152/jn.00524.2019